# Examination of the Changes in Calcium Homeostasis in the Delayed Antiarrhythmic Effect of Sodium Nitrite

**DOI:** 10.3390/ijms20225687

**Published:** 2019-11-13

**Authors:** Vivien Demeter-Haludka, Mária Kovács, János Prorok, Norbert Nagy, András Varró, Ágnes Végh

**Affiliations:** 1Department of Pharmacology and Pharmacotherapy, University of Szeged, H-6721 Szeged, Hungary; demeter-haludka.vivien@med.u-szeged.hu (V.D.-H.); prorok.janos@med.u-szeged.hu (J.P.); nagy.norbert@med.u-szeged.hu (N.N.); varro.andras@med.u-szeged.hu (A.V.); vegh.agnes@med.u-szeged.hu (Á.V.); 2MTA-SZTE Research Group of Cardiovascular Pharmacology, H-6721 Szeged, Hungary

**Keywords:** ischemia/reperfusion, arrhythmia, sodium nitrite, calcium transient, mitochondrial respiration

## Abstract

We have evidence that the intravenous infusion of sodium nitrite (NaNO_2_) results in an antiarrhythmic effect when given 24 h prior to an ischemia and reperfusion (I/R) insult in anaesthetized dogs. This protection was associated with the reduction of reactive oxygen species resulting from I/R through the attenuation of mitochondrial respiration. Here, we examined whether the changes in calcium, which also contributes to arrhythmia generation, play a role in the NaNO_2_-induced effect. On the first day, 30 anaesthetized dogs were treated either with saline or NaNO_2_ (0.2 µmol/kg/min) for 20 min. Some animals were subjected to a 25 min LAD (anterior descending branch of the left coronary artery) occlusion and 2 min reperfusion (I/R = 4; NaNO_2_-I/R = 6), or the heart was removed 24 h later. We have shown that nitrite prevented the I/R-induced increase in cellular and mitochondrial calcium deposits. During simulated I/R, the amplitude of the calcium transient and the diastolic calcium level were significantly lower in the nitrite-treated hearts and the ERP (effective refractory period) fraction of the action potential was significantly increased. Furthermore, nitrite also enhanced the mitochondrial respiratory response and prevented the MPTPT opening during calcium overload. These results suggest that nitrite can reduce the harmful consequences of calcium overload, perhaps directly by modulating ion channels or indirectly by reducing the mitochondrial ROS (reactive oxygen species) production.

## 1. Introduction

Severe ventricular arrhythmias resulting from an acute ischemia and reperfusion (I/R) insult are the main cause of sudden cardiac death. We have previous evidence that inorganic nitrite administered immediately prior to a 25 min coronary artery occlusion, or during the occlusion period but prior to reperfusion, markedly attenuated these ischemia-induced arrhythmias and increased survival in anaesthetized dogs. For example, infusing sodium nitrite 10 min prior to the re-opening of the occluded left anterior descending coronary artery increased survival to 94%, compared with the saline-infused controls, where no dog survived the similar period of ischemia and reperfusion insult [1]. We have also pointed out that this protective effect of nitrite involves *S*-nitrosylation and *S*-glutathionylation mechanisms [1].

We also have evidence that sodium nitrite administered 24 h before the coronary artery occlusion and reperfusion provides a significant protection against the arrhythmias [2]. This marked delayed antiarrhythmic effect of nitrite has been found to be associated with an attenuation of mitochondrial reactive oxygen species (ROS) formation resulting from the depressed mitochondrial respiration by nitrite [3].

There is increasing evidence of the interplay between calcium and oxidative stress products; ROS can modulate calcium release and uptake by modifying calcium channel proteins and transporters, whereas calcium can regulate the mechanisms associated with oxidative stress [4,5]. For example, a significant burst of ROS occurs during reperfusion of the ischemic heart [6,7], resulting in changes in the cellular redox environment. Since the ion channels and transporters, including the calcium channels and transporters, are redox sensitive, accumulation of ROS through the redox modulation of these components of calcium regulation is directly responsible for the I/R-induced calcium overload [8].

The third key player in this scenario is nitric oxide (NO), which can also modulate the redox status of the intracellular milieu, and in most cases, oppose the oxidative effects of ROS via a mechanism termed post-translational S-nitrosylation [9]. However, NO modulates the activity of cardiac ion channels via other mechanisms known as cGMP-dependent signalling pathways, in which following the stimulation of soluble guanylate cyclase and the resulting increase in cGMP, the activation of cGMP-dependent protein kinases and cGMP-regulated phosphodiesterases play a role [10]. In one seminal paper [11], it was proposed that the modulatory role of nitric oxide on cardiac channels, regardless of the mechanism, represents a cardioprotective signalling pathway that lowers the threshold for arrhythmogenesis.

Since both calcium and oxidative stress products are implicated in the generation of severe ventricular arrhythmias resulting from ischemia and reperfusion [5], we designed studies in which the effect of sodium nitrite on calcium homeostasis was examined in both in vivo and in vitro experiments. Thus, in dogs subjected to I/R 24 h after nitrite infusion, the changes in the intracellular and mitochondrial calcium content (calcium deposits) were determined. Furthermore, in mitochondria isolated from dogs infused with nitrite without I/R, the resistance of mitochondrial respiration was also examined in response to calcium overload. Moreover, in samples taken from these nitrite-treated dogs, the changes in the action potential parameters and in the calcium transients were assessed. We also investigated the changes in mitochondrial respiration 24 h after nitrite administration, both in the presence and absence of I/R challenge.

We found that sodium nitrite significantly attenuated the I/R-induced calcium overload (the number of calcium deposits, the amplitude of the calcium transient, and the diastolic calcium level), increased the tolerance of mitochondria to calcium load, and depressed mitochondrial respiration, resulting in a reduction in ROS formation. We propose that these effects on calcium handling may also contribute to the delayed antiarrhythmic effect of nitrite.

## 2. Results

### 2.1. The Effect of Sodium Nitrite on the Intracellular and Mitochondrial Calcium Content

The cellular and mitochondrial calcium content was assessed by the determination of the number of calcium deposits using transmission electron microscopy (Figure 1). The observations were performed in three layers of the cell, namely subsarcolemmal (SM), intermyofibrillar (IM), and perinuclear (PN) regions, and the results are illustrated in original representative images (Figure 1A). Following quantitative analysis, the data were also summarized in bar graphs (Figure 1B). Compared to the sham-operated controls, in samples taken from dogs subjected to a 25 min period of I/R, the number of calcium deposits within the cell and in the mitochondria was significantly increased in all examined regions. The administration of nitrite 24 h prior to the ischemia and the reperfusion insult itself did not modify the number of calcium deposits, however, under ischemic conditions it significantly reduced the I/R-induced intracellular and mitochondrial calcium overload.

### 2.2. The Effect of Sodium Nitrite on the Action Potential (AP) Parameters

This was assessed in multicellular papillary muscle isolated from control and nitrite-treated dogs, and 24 h later subjected to 25 min simulated ischemia and 10 min reperfusion. The results are summarized in Table 1. There were no significant changes in the AP parameters following a 25 min ischemia. The administration of nitrite also did not modify the measured AP potential parameters.

In the control group, ischemia resulted in a significant shortening in APD90 and a reduction in the magnitude of the AP upstroke (AMP) and velocity (Vmax), while the resting membrane potential was slightly depolarized but it remained within the experimental variance. Following reperfusion, most of the AP parameters returned to the normal value. In cells subjected to NaNO_2_ 24 h earlier, the 25 min ischemia resulted in similar changes in most of the AP parameters, except that the decrease in APD90 and Vmax was less pronounced and the ERP/APD ratio was markedly increased during ischemia. In this group, reperfusion also resulted in a full recovery of the AP parameters.

### 2.3. The Effect of Sodium Nitrite on the Calcium Transients

The intracellular calcium changes in cardiac myocytes isolated from control and nitrite-treated dogs during a period of 18 min ischemia and 10 min reperfusion were assessed by changes in the amplitude (Figure 2A), the level of the diastolic calcium (Figure 2B), the time-to-peak value (Figure 2C), and the half-time of relaxation (Figure 2D) of the calcium transient (Figure 2). Compared to the sham controls (not subjected to ischemia), in the ischemic myocytes there was a significant decrease in the amplitude of the calcium transient, which was further reduced in cells obtained from dogs given sodium nitrite 24 h previously (Figure 2A). Following reperfusion, a tendency for restoration of the amplitude of the calcium transient could only be seen in cells of the nitrite-treated dogs (Figure 2A). Furthermore, compared with the sham controls, the ischemia resulted in almost similar increase in the diastolic calcium levels both in the control and the nitrite-treated cells; however, only in cells isolated from the nitrite-treated dogs was the diastolic calcium significantly reduced on reperfusion (Figure 2B). The administration of nitrite resulted in a further significant prolongation of the time-to-peak value in cells that underwent ischemia (Figure 2C), whereas the half relaxation time was not changed compared to the ischemic controls (Figure 2D).

### 2.4. The Effect of Sodium Nitrite on the Tolerance of Mitochondria to Calcium Overload

In order to examine the sensitivity of the mitochondrial permeability transition pore (MPTP) opening to calcium overload, changes in the complex I (CI) and complex II (CII)-dependent mitochondrial calcium retention capacity were determined by fluorescence in cells isolated from control and sodium-nitrite-infused dog hearts with increasing concentrations of calcium chloride (Figure 3). In control mitochondria, the step-by-step increase of the calcium chloride concentration from 30 to 180 µM damaged the mitochondria (Figure 3A,B) after the fourth or fifth injection (average of calcium peaks, CI: 4.7 ± 0.4 and CII: 5.4 ± 0.8, Figure 3C,D). In contrast, the tolerance of the mitochondria to the consecutive calcium injections was significantly increased in mitochondria that had been obtained from dog hearts treated with sodium nitrite 24 h prior. In this case, the CI-dependent calcium uptake was stopped after the 240 µM concentration of calcium chloride (average of calcium peaks 6.5 ± 0.7), whereas the CII-dependent mitochondrial calcium uptake was somewhat more resistant to the calcium overload, where some fluorescent signals could still be observed after the injection of 300 µM calcium (average of calcium peaks 8.5 ± 0.9).

### 2.5. The Effect of Sodium Nitrite on the Mitochondrial Respiratory Parameters

This was examined in mitochondria obtained from dogs given either saline (control) or sodium nitrite 24 h prior. Compared with controls, sodium nitrite significantly reduced both the CI- and CII-dependent OXPHOS and ETS without substantially affecting the RCR and the P/E coupling ratio (Figure 4).

### 2.6. The Effect of Sodium Nitrite on the Tolerance of Mitochondrial Respiration to Calcium Overload

The tolerance of CI- and CII-dependent mitochondrial respiration to increasing concentrations of calcium was determined using a Clarke-type oxygen electrode, and the results are illustrated in Figure 5. In response to the consecutive calcium injections, in samples taken from the control dog hearts an increase was observed at the beginning, but then at the higher calcium concentrations a decrease occurred in both CI- and CII-dependent mitochondrial oxygen consumption. In contrast, in mitochondria prepared from the nitrite-treated dogs, the mitochondrial respiration was preserved, even at the higher calcium concentrations, and it was significantly higher than in the controls (e.g., Ca-6 for CI and Ca-4–6 for CII).

## 3. Discussion

This study aimed to examine whether changes in calcium homeostasis play a role in the delayed antiarrhythmic effect of sodium nitrite. To raise this question was obvious, since our previous studies showed that the administration of sodium nitrite markedly reduces the severity of ischemia and reperfusion-induced ventricular arrhythmias 24 h later and significantly increases the rate of survival [2]. It is well established that the occurrence of these I/R-induced severe ventricular arrhythmias are largely dependent on the generation of reactive oxygen species (ROS) and changes in calcium homeostasis [6,7,12]. We have previous evidence that sodium nitrite attenuates ROS formation by suppressing mitochondrial respiration, most probably acting on the respiratory complexes (mainly complex I) and on the phosphorylation system [3], but we had no information on whether nitrite would modify the ischemia and reperfusion-induced calcium overload involved in arrhythmia suppression. In addition, there is increasing evidence of interplay between ROS and calcium as regards the regulation of various intracellular and mitochondrial calcium signalling pathways under both intact and pathologic conditions [5,8]. This interaction is particularly valid in pathologic situations, where impairment of calcium handling and oxidative stress can reinforce each other, leading to impaired excitation–contraction coupling [13] and severe cardiac arrhythmias [14].

The third factor involved is nitric oxide (NO), a ubiquitous cellular messenger that is involved in the regulation of many cardiac functions, including the modulation of the activity of cardiac ion channels [11]. We have evidence that NO exerts antiarrhythmic effect under conditions of ischemia and reperfusion [15,16]. This highly diffusible gas molecule spreads rapidly from the synthesis site, and as a free radical readily reacts with other species, such as oxygen and superoxide [17]. Nitric oxide exerts its biological effects through two main mechanisms: through cGMP-dependent and cGMP-independent signalling pathways. The first mechanism includes the NO-induced activation of soluble guanylate cyclase and the resulting increase in cGMP, which modulates the activity of cGMP-dependent kinases (e.g., PKG) and cGMP-regulated phosphodiesterases [18]. We have strong evidence that the antiarrhythmic effect of ischemic preconditioning involves the cGMP-dependent nitric oxide pathway [19]. The second cGMP-independent regulatory mechanism is the S-nitrosylation process, during which NO modifies the thiol groups of proteins to form S-nitrosothiol. S-nitrosylation is a ubiquitous post-translational modifying system by which NO modulates the activity of several cardiac channels [9]. Our own studies revealed that the acute antiarrhythmic effect of sodium nitrite is accomplished, at least in part, through protein S-nitrosylation and glutathionylation [1].

The present study, which aimed to examine the role of calcium in the delayed antiarrhythmic effect of sodium nitrite under various in vivo and ex vivo conditions, revealed that nitrite modifies the ischemia reperfusion-induced calcium movements, even 24 h after its administration.

First, we showed that compared to the non-ischemic controls, the 18 min of ischemia significantly reduced the amplitude of the calcium transient and increased the diastolic calcium level and the time to the peak value without substantially modifying the half relaxation time. These indices of intracellular calcium content were different in the nitrite administered heart preparations (i.e., the amplitude of the calcium transient in the nitrite-treated dogs was significantly less than in the ischemic controls, and the time to reach the peak value was also prolonged in samples from the nitrite-treated dogs). Further marked differences between the two groups could be observed on reperfusion. In general, in the presence of nitrite in the preparations showed a better return to the baseline values following ischemia than the untreated controls. For example, following reperfusion in the nitrite-treated preparations there was a better tendency to return to the normal value for the calcium transient amplitude and time-to-peak value than in the ischemic controls. More importantly, compared with the ischemic myocytes, in the nitrite-infused samples the diastolic calcium level was significantly less than in either the ischemic controls or even the initial baseline value. These results suggest that nitrite is able to attenuate the I/R-induced calcium overload, even 24 h after its administration, an effect which certainly may play a role in the antiarrhythmic effect of sodium nitrite.

These alterations in calcium movements induced by nitrite were without substantial changes in the AP parameters. We showed that sodium nitrite infused in dogs 24 h prior to simulated ischemia did not significantly influence the various action potential parameters compared to the untreated controls, except that in the presence of nitrite the ERP/APD was significantly prolonged. This indicates that nitrite prolongs the ERP duration, which might have an antiarrhythmic effect.

The attenuation of the effects of the I/R-induced calcium overload by nitrite is also supported by our TEM investigations, which clearly showed a reduced number of calcium deposits in both the intracellular space and the mitochondria in the nitrite-treated samples compared to the control ones. Although the administration of nitrite alone (without I/R) did not affect the calcium content, it significantly reduced the number of calcium deposits during I/R in almost all examined cell regions.

However, the administration of nitrite significantly depressed the CI- and CII-dependent mitochondrial respiration (OXPHOS) 24 h later in both the absence (Figure 4) and the presence of the I/R insult [3]. As we have proposed previously, a certain extent of reduction in mitochondrial respiration by nitrite would inhibit the formation of ROS (similar to that may occur after the use of an uncoupler) [20], most probably affecting the respiratory complexes and the phosphorylation system [3]. Although the precise mechanism by which nitrite affects the respiratory complexes is still not elucidated, it is proposed that nitrite (NO) *S*-nitrosylates those respiratory complexes that are responsible for ROS formation [21].

It is certain that nitrite increased the tolerance of the mitochondria to calcium overload. Mitochondria isolated from dog hearts given sodium nitrite 24 h previously showed better tolerance to the increasing concentrations of calcium chloride than the untreated controls. Thus, the nitrite-treated mitochondria showed an increase in calcium retention capacity and the mitochondrial respiration stopped at higher calcium concentrations than the control mitochondria. The results suggest that sodium nitrite delays the deterioration of mitochondrial function under conditions of calcium overload, most probably by delaying the opening of MPTP [22] and attenuating the harmful consequences of calcium overload. This is particularly important during reperfusion, when dramatic changes occur in calcium homeostasis and which have substantial roles in the generation of fatal ventricular arrhythmias [23,24].

It has been well established that there is tight interaction between calcium and ROS; accumulation of ROS leads to calcium overload [8,12], and vice versa an increase in intracellular calcium deteriorates cellular functions, including mitochondrial respiration and ROS production [23]. This self-sustaining process under pathologic conditions is inevitably responsible for cell death and for the generation of severe ventricular arrhythmias. Changes in the redox environment resulting from increased ROS production due to I/R favor the redox modification of proteins and lead to structural and functional alterations of ion regulatory proteins, including the calcium channels, pumps, and transporters [4,25]. For example, ROS and reagents that oxidize thiols stimulate cardiac ryanodine receptor (RyR) channels [26], resulting in calcium release from the sarcoplasmic reticulum (SR), and can also enhance calcium entry via the reverse mode of NCX [27]. On the other hand, ROS and oxidizing agents inhibit the SR Ca^2+^ ATPase (SERCA) that results in the inhibition of calcium uptake into the SR [8,28]. These mechanisms are the main ones that are responsible for the increase of intracellular calcium and contribute to cardiac calcium overload under conditions of I/R [29]. In contrast, the reducing conditions and substances, such as NO, inhibit RyR function [30,31] and protect SERCA pump activity from inhibition [32]; both mechanisms can potentially reduce the harmful consequences of calcium overload. Although the precise mechanisms by which NO modulates the gating properties of the cardiac calcium transport channels are still not well understood and the results are rather controversial, there is some evidence for the involvement of both cGMP-dependent and cGMP-independent pathways [33]. For example, RyR activity is modulated by NO via both mechanisms [34,35], depending on the concentration and cytosolic environment [8]. Thus, it has been shown that the endogenously produced NO decreases the open probability of the RyR [30], whereas the NO donor SIN-1 activates the channel by thiol oxidation [36]. In contrast, the SERCA calcium pump activity has not been found to be influenced by S-nitrosylation of cysteine residues directly [37], although there is evidence that peroxynitrite increases the activity of SERCA [32].

Since the present study did not examine the mechanisms by which nitrite regulates the activity of the various calcium transport channels, we have indirect evidence that nitrite affects calcium homeostasis, and this effect may have a role in the delayed antiarrhythmic effect of sodium nitrite. Thus, we propose that nitrite reduces calcium overload under conditions of I/R. This effect may result from the fact that nitrite depresses mitochondrial respiration and attenuates ROS formation during I/R [3]. The lessened ROS formation could indirectly affect the calcium transport by providing a more moderate oxidative milieu under ischemic conditions in the vicinity of the calcium pumps and transporters. The attenuation of the oxidative stress by nitrite, thus, indirectly affects the calcium release and uptake (or outflow) systems and results in reduced calcium overload. Results showed that nitrite significantly reduced the number of calcium deposits, the amplitude of the calcium transient, and the diastolic calcium level under conditions of I/R. Nevertheless, a direct modulatory effect of nitrite (NO) on the calcium channels and transporters cannot be excluded; this, however, has not been examined in the present study. We believe the primary effect of nitrite is on the mitochondria. Nitrite preserves mitochondrial structure and function [3] paradoxically by reducing mitochondrial respiration to an extent that inhibits ROS formation. It is clear from our results that nitrite increases the tolerance of the mitochondria to calcium overload. This is indicated by an increase in calcium retention capacity and the later collapse of mitochondrial respiration following nitrite administration. We propose that the opening of the mitochondrial permeability transition pores and the cessation of mitochondrial respiration occur at higher calcium concentrations in the nitrite-treated hearts than in the untreated controls, which could certainly be associated with the lessened production of ROS by nitrite.

### Limitations of the Study

The primary aim of this study was to examine whether changes in calcium homeostasis play a role in the delayed antiarrhythmic effect of sodium nitrite. The results showed that nitrite attenuates both the cellular and mitochondrial calcium overload during I/R, suggesting a role of calcium in the delayed antiarrhythmic effect of sodium nitrite. Since we have shown previously that changes in ROS production under I/R also plays a major role in the cardioprotective effect of sodium nitrite, in the current study these experiments were not repeated. For the discussion, we used this information and the current data to point out the role of the interaction of ROS and calcium in the antiarrhythmic effect of nitrite.

In the literature, there is increasing evidence that nitrite directly effects calcium channels and transporters by *S*-nitrosylation, and thereby modifies myocardial function. The limitation of our study is that this direct effect of nitrite (*S*-nitrosylation) has not been examined in the current paper.

We think a further limitation of the study might be the limited number of dogs for the measurements of some parameters. However, during the design of this study we had to consider the high cost of the dog studies and the harm-to-benefit ratio outlined by the ethical committee for the use of experimental animals. Therefore, for the explanation of some effects, we used the information obtained from our previous works.

## 4. Material and Methods

### 4.1. Ethics

Experiments were carried out in accordance with the Hungarian law 40/2013 (II. 14.) and were supervised and approved by the Ethical Committee for the Protection of Animals in Research of University of Szeged and the Csongrád County Governmental Office for Food Safety and Animal Health. Approval number: XIII./4657/2016, (CSI/01/4657-6/2016; 23/11/2016).

### 4.2. Animals and Housing

Thirty adult dogs (eighteen males and twelve females) with mean body weight of 14 ± 2 kg were used. The dogs were housed in an environmentally controlled room for two weeks before the experiments, where two animals were kept in one pen, separated by sexes. The temperature was set to 10–20 °C, humidity 40–70%, and lighting 12 h per day. The dogs were fed a standard diet and ad libitum access to water. The food was withdrawn 24 h before the anesthesia.

### 4.3. In Vivo Measurements

#### 4.3.1. Surgical Procedures

The surgical interventions were performed as described previously [38,39]. On day one, the dogs were lightly anaesthetized with sodium pentobarbitone (30 mg/kg, i.v., Euthasol 40%, Produlab Pharma B.V., Raamsdonksveer, The Netherlands). The jugular vein was prepared and either saline or NaNO_2_ was administered in intravenous infusion through a polyethylene catheter. Arterial blood pressure was measured from the left carotid artery by a Millar tip catheter (F5, Millar Instruments Inc., Colorado Springs, CO, USA). Heart rate (HR) was assessed from chest lead II electrocardiogram.

Twenty-four hours later, the dogs were re-anaesthetized with a bolus injection of sodium pentobarbitone (30 mg/kg, i.v.; Euthasol 40%, Produlab Pharma B.V., Raamsdonksveer, The Netherlands) and the anesthesia was maintained with the mixture of α-chloralose and urethane (60 and 200 mg/kg, i.v.; Sigma, St. Louis, USA). The depth of the anesthesia was monitored by the examination of reflexes. The dogs were intubated and ventilated with room air using a Harvard respirator (Harvard Apparatus, Holliston, MA, USA) and the blood gas values were monitored [38]. Body temperature was maintained at 37 ± 0.5 °C. The right femoral artery was prepared, and a catheter was introduced to measure the systolic and diastolic arterial blood pressure. Left ventricular (LV) systolic (LVSP) and end-diastolic pressure (LVEDP), as well as the LV positive and negative dP/dt_max_, were measured using a Miller tip catheter (5F, Millar Instruments Inc., Colorado Springs, USA) positioned into the left ventricle.

After thoracotomy at the fifth intercostal space, the anterior descending branch of the left coronary artery (LAD) was prepared, and myocardial ischemia was induced by a 25 min occlusion of the LAD, followed by a 2 min period of reperfusion [38]. At the end of the experiments, the animals were euthanized with a bolus sodium pentobarbitone (150 mg/kg, i.v.). All parameters were recorded on a Plugsys Hemodynamic Apparatus (Hugo Sachs Electronik, March, Germany) and evaluated by LabChart 7 software (AD Instruments, Bella Vista, Australia).

#### 4.3.2. In Vivo Experimental Protocol

Thirty dogs of both sexes were used and divided into four experimental groups. In two groups, either saline (I/R control; *n* = 4) or sodium nitrite (NaNO_2_, Merck, Darmstadt, Germany, 0.2 µmol/kg/min) was administered intravenously for 20 min (NaNO_2_ + I/R, *n* = 6), and 24 h later these dogs were subjected to a 25 min ischemia followed by a rapid reperfusion. In the other two groups, either saline (SC; *n* = 10) or sodium nitrite (NaNO_2_; *n* = 10) was infused, and 24 h later the experiments were terminated without subjecting the animals to I/R. At the end of the experiments, the hearts were stopped by an excess of anesthetic and tissue samples from the area supplied by the LAD were taken for in vitro analyses. The samples were immediately used for mitochondrial measurements and for cardiomyocyte isolation, respectively.

### 4.4. In Vitro Measurements

#### 4.4.1. Measurement of the Cellular and Mitochondrial Calcium Content

This was performed by transmission electron microscopy (TEM). After the heart was removed, tissue blocks (1 mm^3^) were taken from the area supplied by the LAD from both the ischemic and non-ischemic groups and fixed in glutaraldehyde (3%; Polysciences, Hirschberg an der Bergstrasse, Germany) and potassium oxalate solution (90 mM; Sigma, St. Louis, MS, USA) overnight. The samples were rinsed in sucrose (7.5%; Sigma, St. Louis, MS, USA) and potassium oxalate (90 mM; Merck, Darmstadt, Germany), and post-fixed in potassium pyroantimonate (2%; Merck, Germany) and osmic acid (1%; Sigma, St. Louis, MS, USA). After rinsing in distilled water, the tissue blocks were dehydrated using graded series of alcohol, embedded in epoxy resin (Durcupan ACM, Sigma, St. Louis, MS, USA), and polymerized at 56 °C for two days. After polymerization, 0.3 µm semithin and 50 nm ultrathin sections were cut with ultra-microtome (Ultracut UCT, Leica, Wetzlar, Germany). The semithin sections were screened under light microscope. The ultrathin sections were contrasted with uranyl acetate (Electron Microscopy Sciences, Hatfield, UK) [40] and lead citrate (Electron Microscopy Sciences, Hatfield, UK) [41]. Transmission electron microscopy (CEM 902, Zeiss, Oberkochen, Germany) was used in conventional transmission mode (80 keV), using a Spot RT 14.0 CCD camera (Diagnostic Instruments, Sterling Heights, MI, USA) at 12,000× magnifications. From each ultrathin tissue section, five images were evaluated and averaged. The mitochondria were segmented with ImageJ 2 (FIJI; NIH, Bethesda, MD, USA). The precipitation of calcium by the fixative resulted in electron-dense calcium deposits. The calcium content was calculated by counting the number of electron-dense calcium deposits, and the results were expressed as the percentage of grid points overlaying either on the whole cell or on the mitochondria. Data obtained from four images in each animal were averaged, and the results obtained from the individual dogs within a certain group were also averaged.

#### 4.4.2. Determination of the Mitochondrial Respiration and Its Tolerance to Calcium Overload

Left ventricle tissue segments were taken from the myocardial region supplied by the LAD and perfused with phosphate-buffered saline solution through the LAD immediately after the hearts were removed. Tissue blocks of 750 mg were excised and homogenized in an isotonic sucrose medium containing trypsin, and the mitochondria were separated by centrifugation. The protein content of the isolated mitochondria was determined by the Bradford protein assay and 0.1 mg of isolated mitochondria was used in mir05 buffer [42] for the respiratory measurements.

A Clarke-type oxygen electrode (Strathkelvin 782 Oxygen Systems, Strathkelvin, North Lanarkshire, Scotland) was used to measure the mitochondrial respiration according to the SUIT protocol [42]. Mitochondrial complex I (CI) and complex II (CII) respiration were induced by either glutamate (10 mM) and malate (1 mM) or rotenone (0.5 µM) and succinate (10 mM). To measure the oxidative phosphorylation (OXPHOS), ADP (5 mM) was injected into the chamber. The state 4 respiration was assessed by adding oligomycin (2.5 µM). The uncoupling of respiration was measured by the injection of carbonyl-cyanide-p-(trifluoro-methoxy) phenyl-hydrazone (ETS, FCCP 0.5 µM). Antimycin A (5 µM) was used to measure the residual oxygen consumption. From the measured parameters, respiratory control ratio (RCR = OXPHOS/state 4) and P/E (OXPHOS/ETS) ratio were calculated. All substrates and inhibitors were purchased from Sigma (St. Louis, USA).

In samples where the effect of calcium overload on mitochondrial respiration was assessed, consecutive injections of 30 µM calcium chloride (CaCl_2_) [23,43] were added to the chamber at one minute intervals. In each experiment, 0.1 mg of isolated mitochondria was used.

#### 4.4.3. Assessment of the Calcium Retention Capacity (CRC)

In order to measure the susceptibility of mitochondrial permeability transition pore (MPTP) opening to calcium overload, 0.25 mg of isolated mitochondria was loaded with Calcium-Green 5N (2 µM, Invitrogen, Waltham, USA) in 0.2 mL TC buffer (150 mM sucrose, 50 mM KCl, 2 mM KH_2_PO_4_, 20 mM Tris-HCl, pH 7.4 at 37 °C) [44]. Mitochondrial CI- and CII-dependent respiration was induced as described above. Calcium overload was induced by step-by-step injections of 30 µM CaCl_2_ until a sudden increase in the extra-mitochondrial fluorescence signal occurred, indicating the opening of the MPTP. The fluorescent signal emitted by Calcium-Green 5N was measured using a plate reader (λex/em = 500/530 nm; FLUOstar OPTIMA, BMG LABTECH, Ortenberg, Germany). Data were expressed as relative fluorescent units (RFU). Cyclosporine A (1 µM, Novartis, Basel, Switzerland), a pore desensitizer, was used as a positive control.

#### 4.4.4. The Assessment of Calcium Transients

The isolation of cardiomyocytes was performed as described previously [45]. In brief, cardiac myocytes were isolated from the left ventricle, containing an arterial branch through which the segment was perfused on a Langendorff apparatus with Tyrode′s solutions in the following sequence: (1) normal Tyrode′s solution for 10 min, (2) Ca^2+^-free Tyrode solution for 10 min, and (3) Ca^2+^-free Tyrode solution containing collagenase (Worthington type II, 0.66 mg/mL). Protease (Sigma type XIV, 0.12 mg/mL) was added to the final perfusate after 15 min and an additional 30 min of digestion was allowed. Cells were stored in Tyrode’s solution (135 mM NaCl, 4.7 mM KCl, 1.2 mM KH_2_PO_4_, 1.2 mM MgSO_4_, 10 mM HEPES, 4.4 mM NaHCO_3_, 10 mM glucose, 1 mM CaCl_2_ (pH 7.2 adjusted with NaOH). All chemicals were purchased from Sigma (St. Louis, USA). The isolated cardiomyocytes were loaded with Fluo-4-AM (1–2 μM, Molecular Probes, Waltham, MA, USA; AM is the membrane permeable acetoxymethyl ester conjugated form of the dye) for 20 min at room temperature. The loaded cells were placed in a low volume imaging chamber (RC47FSLP, Warner Instruments, Hamden, CT, USA), and the cells were then continuously perfused with normal or ischemic Tyrode solution at 37 °C (1 mL/min). Myocytes were stimulated at a constant frequency of 1 Hz through a pair of platinum electrodes using an electronic stimulator (MDE Ltd. EXP-ST02, Walldorf, Germany). The fluorescent measurements were performed on a stage of an inverted fluorescent microscope (IX71, Olympus, Tokyo, Japan) and the signal was recorded by a photomultiplier module (H7828, Hamamatsu, Hamamatsu City, Japan) sampled at 1 kHz. Data acquisition and analysis were performed using the Isosys software (Experimetria Ltd., Budapest, Hungary).

Changes in parameters of Ca transient were characterized by the emitted fluorescence intensity at 535 nm wavelength, following optical signal correction steps for photobleaching and nonspecific background fluorescence. Amplitudes of the [Ca^2+^]_i_ transients were calculated as differences between systolic peak and diastolic values. Diastolic [Ca^2+^]_i_ levels were determined immediately before the trigger stimulus. Systolic [Ca^2+^]_i_ was determined at the peak of the corresponding transient. The rise of peak parameter determines the elapsed time between the end diastolic (or trigger) and maximal peak value. In parallel, the half relaxation time measures the time starting from the maximum peak value until the fluorescence intensity is halved. Background fluorescence levels, recorded several times, were used to correct raw fluorescence data. The [Ca^2+^]_i_ changes were expressed as fluorescence measured over basal unstimulated fluorescence. Time-to-peak and relaxation time were determined in milliseconds and expressed in percentage compared to the baseline values.

#### 4.4.5. Determination of the Action Potential Parameters in the Canine Multicellular Papillary Muscle

Action potentials were recorded at 37 °C from the surface of ventricular papillary muscles using conventional microelectrode techniques. The preparations were mounted in a custom made Plexiglas chamber, allowing continuous super-fusion with O_2_ saturated Krebs–Henseleit solution (118.5 mM NaCl, 4 mM KCl, 2 mM CaCl_2_, 1 mM MgSO_4_, 1.2 mM NaH_2_PO_4_, 25 mM NaHCO_3_, and 10 mM glucose, pH = 7.35 ± 0.05) and stimulated with constant current pulses of 2 ms duration at a rate of 1 Hz through a pair of bipolar platinum electrodes using an electro-stimulator (EX-ST-A2, Experimetria Ltd., Budapest, Hungary). Sharp microelectrodes with tip resistance of 10–20 MΩ filled with 3 M KCl were connected to an amplifier (Biologic Amplifier, model VF 102). Voltage output from the amplifier was sampled using an AD converter (NI 6025, Unisip Ltd., Budapest, Hungary). APD, determined at 90% level of repolarization (APD90), was obtained using Evoke-wave v1.49 (Unisip Ltd., Budapest, Hungary). During the experiments, efforts were made to maintain the same impalement, and when it was dislodged an adjustment was applied. The measurements were only continued if the action potential characteristics of the re-established impalement deviated less than 5% from the previous one.

#### 4.4.6. Experimental Protocol for Simulated Ischemia with Isolated Cardiomyocytes

Cardiomyocytes isolated from the NaNO_2_ and SC groups, detailed above, were used to measure the calcium transient and action potential under conditions of simulated ischemia. In case of the [Ca^2+^]_i_ measurements, an ischemic environment was induced using a low pH, high K^+^, glucose-free, high lactate perfusion solution (123 mM NaCl, 6 mM NaHCO_3_, 0.9 mM NaH_2_PO_4_, 8 mM KCl, 0.5 mM MgSO_4_, 2.5 mM CaCl_2_, and 20 mM lactate), as described elsewhere [46]. The cells were perfused first with normal Tyrode solution for 3 min, and then with ischemic solution for 18 min, followed by a 10 min perfusion period with normal Tyrode solution (reperfusion). During the ischemic period, an oxygen-free gas combination (95% N_2_ +5% CO_2_) was layered over the solution.

For the action potential recordings, ischemia was induced by N_2_-CO_2_ (95% + 5%) gas bubbling for 25 min in the presence of normal Krebs–Henseleit solution. This was replaced for O_2_-CO_2_ bubbling upon reperfusion for 10 min. The recordings were performed before and at the last min of the ischemic period and measurements were also acquired at the 2nd and the 10th min during reperfusion.

### 4.5. Statistical Analysis

Data were expressed as means ± SEM. For the statistical analysis, Student′s t-test or Welch’s ANOVA and Bonferroni–Holm post hoc tests were used. Differences between groups were considered significant at *p* < 0.05. For the electrophysiological and mitochondrial energetics experiments, data obtained from at least two parallel measurements were averaged for each animal and the results obtained from the individual dogs (*n*) within a certain group were also averaged.

## Figures and Tables

**Figure 1 ijms-20-05687-f001:**
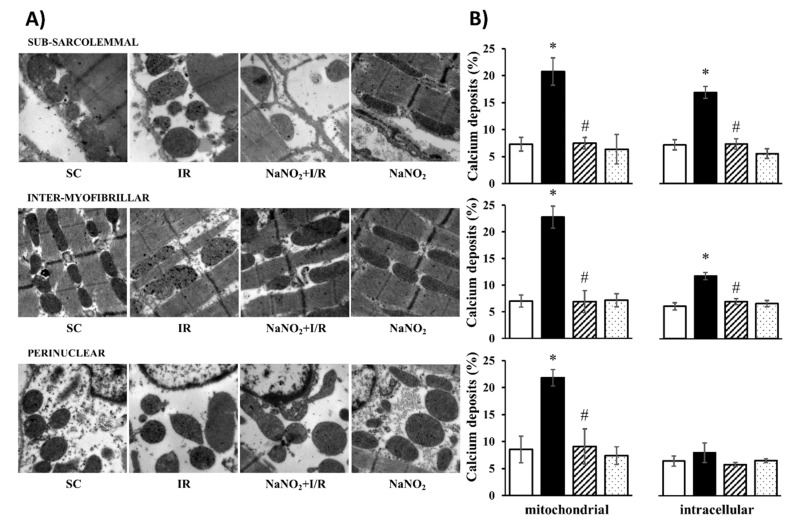
Representative images taken at 12000x magnification (**A**) and their quantitative analysis (**B**) of calcium deposits obtained by transmission electron microscopy (TEM) in the sham-operated (SC, *n* = 4) ischemic control (IR, *n* = 4) and sodium-nitrite-treated groups with (NaNO_2_ + ischemia and reperfusion (I/R), *n* = 4) or without I/R (NaNO_2_, *n* = 3). For each *n*, four or five pictures were evaluated and averaged. The images were taken from the subsarcolemmal, intermyofibrillar, and the perinuclear regions. Data are expressed as means ± SEM. Note: * *p* < 0.05 compared with SC, # *p* < 0.05 compared with IR.

**Figure 2 ijms-20-05687-f002:**
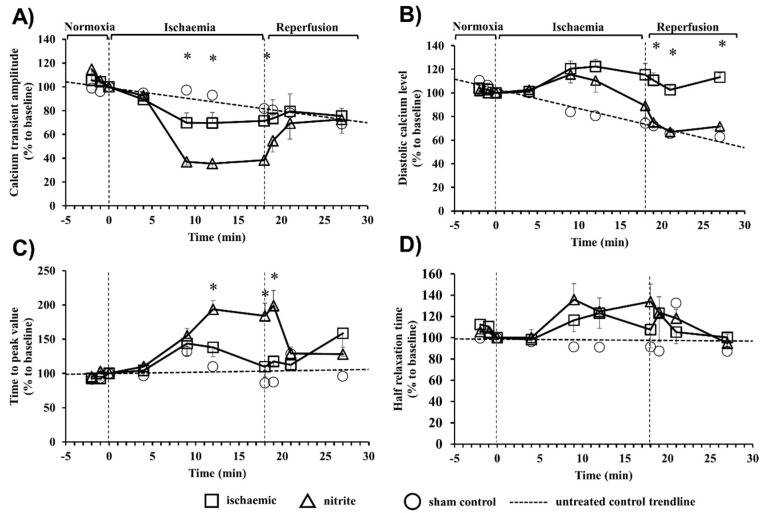
The effect of NaNO_2_ on the intracellular calcium transients in isolated cardiac myocytes. The changes in the Ca transient amplitude (**A**), the diastolic calcium level (**B**), the time-to-peak value (**C**), and in the half-time of relaxation (**D**) are shown. Isolated cardiomyocytes from the sham (*n* = 6) and nitrite-treated dogs (*n* = 8) were subjected to a 18 min ischemia and 10 min of reperfusion protocol. In order to assess the excretion of the fluorescent dye over time, some sham control cells (*n* = 3) were loaded with Fura-4-AM but not subjected to stimulated I/R. A trendline was fitted to these cells in order to measure the loss in fluorescence. Values are means ± SEM. Note: * *p* < 0.05 compared with the control.

**Figure 3 ijms-20-05687-f003:**
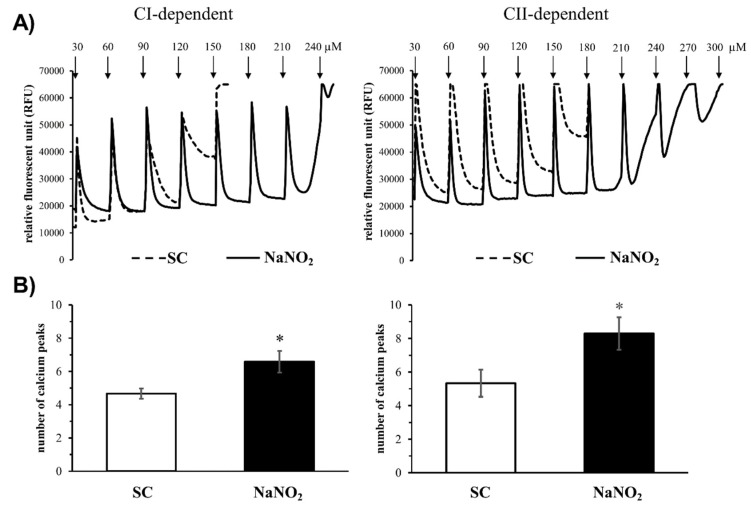
Representative images (**A**) and quantitative data (**B**) of the mitochondrial calcium retention capacity (CRC) of the mitochondria isolated both from the sham and sodium-nitrite-treated groups (SC-CI, *n* = 6; NaNO_2_-CI, *n* = 6; SC-CII, *n* = 6, NaNO_2_-CII, *n* = 5). The calcium overload was stimulated by repeated administration of 30 µM of CaCl_2_ at each measured time point. Values are means ± SEM. Note: * *p* < 0.05 compared with SC.

**Figure 4 ijms-20-05687-f004:**
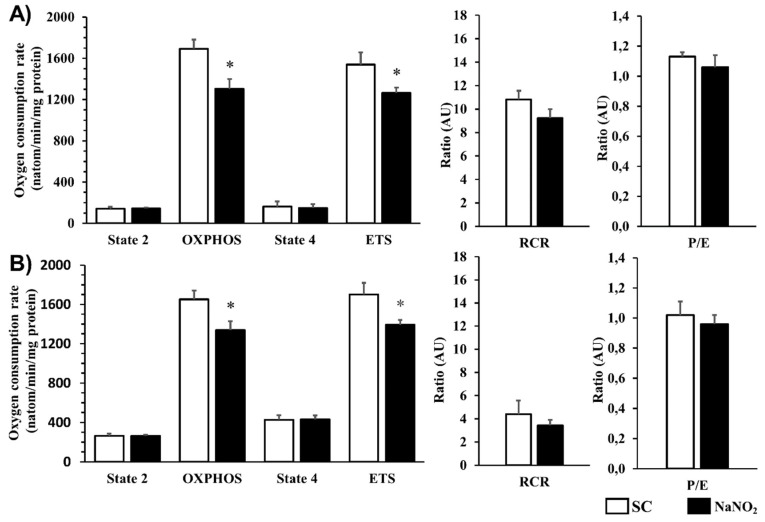
The effect of NaNO_2_ administration on the complex I (CI (**A**))- and complex II (CII (**B**))- dependent basal (state2), ADP-activated active respiration (OXPHOS), and uncoupling of the ETS. The respiratory control ratio (RCR) and the P/E coupling ratio (P/E) were also calculated in mitochondria isolated from the sham and nitrite-treated animals (SC: *n* = 8, NaNO_2_: *n* = 9). Values are means ± SEM. Note: * *p* < 0.05 compared with SC.

**Figure 5 ijms-20-05687-f005:**
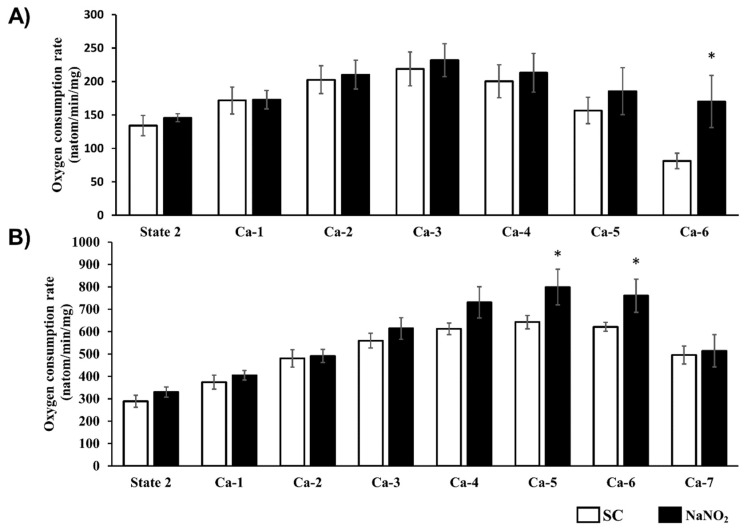
CI (**A**)- and CII (**B**)-dependent respiratory response to calcium overload of mitochondria isolated from the sham-operated (CI, *n* = 6; CII, *n* = 6) and the sodium-nitrite-treated groups (CI, *n* = 11stimulated by repeated administration of 30 µM of CaCl_2_ at each measured time point.

**Table 1 ijms-20-05687-t001:** The effect of NaNO_2_ on the action potential parameters.

Experimental Groups	Time (min)	APD90 (ms)	APD50 (ms)	APD25 (ms)	AMP (mV)	Vmax (*V*/*s*)	RMP (mV)	CT (ms)	ERP (ms)	ERP/APD90
SC	baseline	200 ± 5	166 ± 6	124 ± 7	98 ± 2	184 ± 19	−81 ± 2	4 ± 0.62	206 ± 4	1.02 ± 0.01
	I-25 min	177 ± 3 *	145 ± 3 *	107 ± 6	90 ± 1^*^	121 ± 5 *	−76 ± 3	5 ± 0.25	188 ± 5 *	1.06 ± 0.02
	R-2 min	194 ± 13	153 ± 8	117 ± 5	91 ± 2^*^	138 ± 7	−78 ± 4	5 ± 0.04	193 ± 10	1.01 ± 0.02
	R-10 min	199 ± 9	166 ± 8	123 ± 7	93 ± 4	149 ± 7	−81 ± 4	5 ± 0.26	199 ± 6	0.99 ± 0.02
NaNO_2_	baseline	211 ± 5	174 ± 4	132 ± 4	99 ± 2	177 ± 23	−82 ± 2	4 ± 0.56	220 ± 8	1.03 ± 0.02
	I-25 min	184 ± 11	149 ± 9 *	111 ± 7 *	82 ± 3 *	126 ± 23	−75 ± 2 *	4 ± 0.69	210 ± 14	1.19 ± 0.03 * #
	R-2 min	199 ± 7	157 ± 9	107 ± 11	85 ± 3 *	147 ± 21	−78 ± 3	4 ± 0.62	216 ± 10	1.08 ± 0.05
	R-10 min	205 ± 9	160 ± 7	116 ± 9	94 ± 4	169 ± 19	−83 ± 3	4 ± 0.45	216 ± 5	1.05 ± 0.02

Values are means ± SEM, calculated from *n* = 3 sham controls and *n* = 5 NaNO_2_-treated dogs. Abbreviations: APD 25, 50, 90 = action potential duration 25, 50, 90; AMP = amplitude; Vmax = maximum upstroke velocity (V/s = volt per second); RMP = resting membrane potential; CT = conduction time; ERP = effective refractory period. Student′s t-test. Note: * *p* < 0.05 compared with baseline, # *p* < 0.05 compared with the control group.

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
