# Peer review of "Examination of the Changes in Calcium Homeostasis in the Delayed Antiarrhythmic Effect of Sodium Nitrite"

_ijms, 2019, doi:10.3390/ijms20225687_

Round 1

Reviewer 1 Report

Some points need to be addressed prior to publication:

English should be revised. The title should be affirmative and not a question. The number of dogs used should be well explained. Is there a sample size analysis? Experimentally the manuscript is sound, but the discussion give a large space to ROS based on literature, but no ROS measures are presented in the manuscript. Although several experiments have been already done, some ROS measures as “confirmatory effects” could be included.

Minor:

Page 15, line 357: p should be 0,05.

Author Response

Response to Reviewer 1 Comments

We thank reviewer 1 for his/her valuable comments. We have answered to these questions point by point.

Point 1: English should be revised.

Response 1: This has now been corrected by an English editor; the grammar, spelling and punctuation have been checked. The changes made by the editor are marked with the Microsoft Office track-changes mode in the manuscript.

Point 2: The title should be affirmative and not a question.

Response 2: The title has been changed to the following:

’Examination of the changes in calcium homeostasis in the delayed antiarrhythmic effect of sodium nitrite.’

Point 3: The number of dogs used should be well explained. Is there a sample size analysis?

Response 3: The number of dogs and measurements are indicated in Table1 (only for the use of the referee). We have used for the estimation of the sample size, the PS Power and Sample Size Calculations Software (Version 3.1.6) with the following parameters, α was set to 0.05 and the power to 0.8. In four cases out of six measurements; TEM, CRC, respiration and respiratory response to calcium overload, the number of dogs met the requirements resulted from the Power sample size analysis. However, in CaT and AP measurements, the number of animals was somewhat lower than the required dog numbers. In these cases, we have disregarded to increase the number of controls because we had to consider the high cost of the animals, the ratio of the harm-benefit, further we have already had such previous measurements including in the article of Kormos et al. (Eur J Pharmacol, 2014).

Because in some cases, the low number of dogs do not allow to examine normality, we used as previously (Demeter-Haludka et al. Frontiers in Pharmacol, 2018) the Welch-ANOVA with Bonferroni-Holm post hoc test for the statistical analysis. This does not require normal distribution and identical n numbers.

Point 4: Experimentally the manuscript is sound, but the discussion give a large space to ROS based on literature, but no ROS measures are presented in the manuscript. Although several experiments have been already done, some ROS measures as „confirmatory effects” could be included.

Response 4: As we have declared the main aim of our studies was to determine two main participants playing a role in the mechanism of the generation of I/R induced arrhythmias and in the delayed antiarrhythmic effect of sodium nitrite. One of these is the ROS which role has already been examined and published previously (Demeter-Haludka et al. Frontiers in Pharmacol, 2018). The other key player in the I/R induced arrhythmia generation is the changes in calcium homeostasis which role now has been examined in the current paper. Therefore, in the discussion, we have focused on the interaction between these two players to which nowadays a great importance has been attributed.

Since, in the present study the ROS levels have not been determined, we used those data for the interpretation of this interaction coming from our previous experiments published in Frontiers in Pharmacology (Demeter-Haludka et al. Frontiers in Pharmacol, 2018). We should like to keep the discussion as it stands. However, as referee 2 required we have included a new paragraph in which the limitations of the study have been addressed.

Point 5: Minor: Page15, line357: p should be 0,05

Response 5: This has now been corrected.

Reviewer 2 Report

This manuscript is written well, so I recommend to publish.

In addition, I recommend to add more study limitations in this study.

1) This manuscript is written well, however, this will need to be revised English.
2) About the hypothesis of this study.
Please be more specific.

3) About the title.
Are you adapting to your conclusions and hypotheses?
Please consider again.

4) About setting the sample size
Please indicate specifically.

5) Please consider again the limitations of research.

Author Response

Response to Reviewer 2 Comments

We thank reviewer 2 for his/her valuable comments. We have answered to these questions point by point.

Point 1: This manuscript is written well, however, this will need to be revised English.

Response 1: This has now been corrected by an English editor; the grammar, spelling and punctuation have been checked. The changes made by the editor are marked with the Microsoft Office track-changes mode in the manuscript.

Point 2: About the hypothesis of this study. Please be more specific.

Response 2: We have now modified the introduction and made the hypothesis or the aim more specific. Page2, lines 68-72.

Point 3: About the title: Are you adapting to your conclusions and hypotheses? Please consider again.

Response 3: The title has been changed to the following:

’Examination of the changes in calcium homeostasis in the delayed antiarrhythmic effect of sodium nitrite.’

Point 4: About setting the sample size. Please indicate specifically.

Response 4: The sample size calculations were made, please see our answers given to referee 1.

Point 5: Please consider again the limitations of research.

Response 5: As the referee advised, we have now included a new paragraph to the end of the discussion and titled ‘Limitation of the study’. Page 19, lines 199-215.

Round 2

Reviewer 1 Report

acceptable.